# Short Digital Spatial Memory Test Detects Impairment in Alzheimer’s Disease and Mild Cognitive Impairment

**DOI:** 10.3390/brainsci11101350

**Published:** 2021-10-14

**Authors:** Jackie M. Poos, Ineke J. M. van der Ham, Anna E. Leeuwis, Yolande A. L. Pijnenburg, Wiesje M. van der Flier, Albert Postma

**Affiliations:** 1Alzheimer Center Amsterdam, Department of Neurology, Amsterdam Neuroscience, Amsterdam University Medical Center, Vrije Universiteit Amsterdam, 1007 MB Amsterdam, The Netherlands; a.leeuwis@amsterdamumc.nl (A.E.L.); yal.pijnenburg@amsterdamumc.nl (Y.A.L.P.); wm.vdflier@amsterdamumc.nl (W.M.v.d.F.); 2Department of Neurology, Erasmus MC University Medical Center, 3015 GD Rotterdam, The Netherlands; 3Helmholtz Institute, Experimental Psychology, Utrecht University, 3584 CS Utrecht, The Netherlands; a.postma@uu.nl; 4Institute of Psychology, Health, Medical and Neuropsychology, Leiden University, 2300 RB Leiden, The Netherlands; c.j.m.van.der.ham@fsw.leidenuniv.nl; 5Department of Epidemiology & Biostatistics, Vrije Universiteit Amsterdam, Amsterdam UMC, 1081 HV Amsterdam, The Netherlands

**Keywords:** cognitive dysfunction, memory disorders, spatial navigation, neuropsychology, early diagnosis, dementia

## Abstract

Background: Impairment in navigation abilities and object location memory are often seen in early-stage Alzheimer’s Disease (AD), yet these constructs are not included in standard neuropsychological assessment. We investigated the differential ability of a short digital spatial memory test in mild AD dementia and mild cognitive impairment (MCI). Methods: 21 patients with AD dementia (66.9 ± 6.9; 47% female), 22 patients with MCI (69.6 ± 8.3; 46% female) and 21 patients with subjective cognitive decline (SCD) (62.2 ± 8.9; 48% female) from the Amsterdam Dementia Cohort performed the Object Location Memory Test (OLMT), consisting of a visual perception and memory trial, and the Virtual Tübingen (VT) test, consisting of a scene recognition, route continuation, route ordering and distance comparison task. The correlations with other cognitive domains were examined. Results: Patients with mild AD dementia (Z: −2.51 ± 1.15) and MCI (Z: −1.81 ± 0.92) performed worse than participants with SCD (Z: 0.0 ± 1.0) on the OLMT. Scene recognition and route continuation were equally impaired in patients with AD dementia (Z: −1.14 ± 0.73; Z: −1.44 ± 1.13) and MCI (Z: −1.37 ± 1.25; Z: −1.21 ± 1.07). Route ordering was only impaired in patients with MCI (Z: −0.82 ± 0.78). Weak to moderate correlations were found between route continuation and memory (r(64) = 0.40, *p* < 0.01), and between route ordering and attention (r(64) = 0.33, *p* < 0.01), but not for the OLMT. Conclusion: A short digital spatial memory test battery was able to detect object location memory and navigation impairment in patients with mild AD dementia and MCI, highlighting the value of incorporating such a test battery in standard neuropsychological assessment.

## 1. Introduction

Alzheimer’s disease (AD) is the most common form of dementia and is associated with marked impairment in episodic memory, and subsequent decline in executive functioning, language and visuospatial abilities [1]. The clinical stages of the AD pathological process include a preclinical stage, where subjective cognitive decline (SCD) may already be present [2], a prodromal stage that includes mild cognitive impairment (MCI) [3] and a symptomatic stage that is based on the severity of impairment and can be classified as mild, moderate and severe dementia [4]. The clinical diagnostic outcome measures to detect early stages of AD, including MCI, are currently strongly focused on verbal episodic memory tests [5], whereas the complaints of patients with AD dementia, in particular, often include wandering or getting lost while driving [6,7,8,9]. Given the frequency of spatial memory complaints in the early stages of AD dementia [6,7,8,9], proper evaluation of these processes is urgent but lacking in most standard diagnostic work-up settings.

Spatial memory consists of multiple visuospatial abilities that are a crucial part of our daily functioning. It involves the encoding, storage and retrieval of information about spatial layouts, which enables us to remember the positions of objects in place (object location memory) and to learn and remember routes (spatial navigation) [10]. Both processes have been shown to be related to atrophy of the hippocampus, the parahippocampal gyrus and parietal areas [11,12], areas that are known to show the first signs of neurodegeneration in AD [13]. Indeed, spatial memory impairment is not limited to patients with AD dementia, but can be detected earlier in patients with MCI [14].

However, currently, there are no comprehensive and ecologically valid tests available that are feasible for application in clinical settings [15]. Spatial memory processes such as object location memory and navigation are typically conducted in dynamic and complex environments, making it difficult to perform ecologically valid assessments in a clinical setting [16]. There are validated tabletop visuospatial tests such as the Mental Rotation Test [17], the Location Learning Test [18], the Corsi Block Tapping test [19] and the Money Road Map test [20]. However, these small-scale spatial tests are highly inflexible, measure only a single aspect in the spatial memory spectrum or have proven to be poor predictors of navigational abilities [21] and cognitive decline [22]. Other studies have aimed to simulate the complexity of real-life situations by assessing large-scale spatial navigation in real-world scenarios such as hospital settings (for example, [11,14,23,24,25,26,27,28]) or with innovative tests incorporating advanced virtual reality (VR) paradigms such as the Memory Island Test and the Sea Hero Quest [14,25,26,29,30,31,32,33,34,35,36,37,38]. Multiple studies have proven that real-world scenarios and VR applications appear to be more sensitive in identifying spatial navigation deficits in patients with AD in both the prodromal and symptomatic stage [11,14,25,26,39,40,41,42]. However, these experimental tests using real-world scenarios and/or VR applications are time-consuming (for example, most experimental tests take around 2–3 h to complete) and not feasible for clinical evaluation [15].

In light of this, the aim was to investigate the differential ability of a short digital test battery that is easy to administer and measures the most relevant aspects of spatial memory in a systematic and condensed, time-limited manner in patients with mild AD dementia and MCI. The Object Location memory Test (OLMT) is a computer program with which multiple crucial spatial memory processes such as spatial perception and construction, object and location memory, the binding of objects to locations and metric distance processing can easily be measured [10,43]. The test has been previously used and validated in healthy controls and stroke patients [10,43]. For spatial navigation, we used a virtual reality environment, the Virtual Tübingen (VT) task [12], including tasks to measure landmark recognition, location and path knowledge, in agreement with theories on navigation abilities [12,44]. We performed correlational analyses with six other neuropsychological cognitive domains to assess the relation between the spatial memory tests and well-known cognitive tests that are a part of the standard diagnostic work-up.

## 2. Materials and Methods

### 2.1. Participants

We recruited 21 patients with probable AD dementia, 22 patients with MCI and 21 participants with subjective cognitive decline (SCD) from the Amsterdam Dementia Cohort [45,46]. The diagnoses were established at a multidisciplinary consensus meeting of the Alzheimer Center Amsterdam, involving experienced neurologists, neuroradiologists, psychiatrists, neuropsychologists and a care consultant, according to established diagnostic criteria for MCI [3] and AD dementia [4]. Clinical diagnosis was based on a standardized clinical assessment consisting of a medical history, family history, physical examination, neuropsychological assessment, EEG and MR imaging of the brain [45,46], according to NIA–AA criteria [3,4]. Patients with other neurological/psychiatric diseases, traumatic brain injury and/or major vascular damage were excluded. Patients were labeled as having SCD when cognitive and laboratory investigations were normal compared to normative data and the criteria for MCI, dementia or any other neurological/psychiatric disorder known to cause cognitive complaints were not met. All patients received an information letter before their regular follow-up appointment and were recruited and tested directly afterwards. Exclusion criteria were ≤18 on the Mini-Mental State Examination (MMSE) or >1 on the Clinical Dementia Rating scale (CDR), thereby excluding patients with moderate or severe AD dementia.

### 2.2. Experimental Task Design and Procedure

Participants were comfortably seated in a quiet room in front of a touchscreen monitor (19 inch) at a distance of ±60 cm. Data from the experiment were collected by registering touch presses on the monitor. The OLMT [47] consisted of four trials of which two were practice trials. The perception trial was to measure perceptual abilities, to control for possible perceptual impairments. Two 14.5 × 14.5 cm squares were shown on a touchscreen monitor, with ten randomly placed objects in the left square. Each participant was given the following instruction: “You will see two square frames, with the left containing ten objects. On the right there is an empty square with ten identical objects on top. You have to reconstruct the object array in the right frame, making sure the positions match those in the left frame”. For the memory trial, one 14.5 cm square was shown for 30 s with ten different objects. Participants were given the following instruction: “You will see a square containing objects for 30 s. You have to remember the locations of these objects as accurately as possible. After the object disappear, you will see an empty square, and you have to relocate the objects to the correct locations”. For the OLMT, absolute error rates and best-fit scores were calculated. The absolute error is the mean absolute distance in millimeters between the original and the relocated positions of the objects. For the best-fit score, all possible configurations between the original and the relocated positions were computed and the fit that had the smallest error rate was considered to be the best-fitting configuration [47]. In Figure 1, the perception and memory trial of the OLMT can be seen.

The VT test [12] consisted of a study phase with a walking route in a virtual environment and four tests: scene recognition, route continuation, route ordering and distance comparison. Scene recognition was tested with twelve static images of which six were taken from the study phase movie, and six distractors. Route continuation involved six images from six crossing points on the route, where participants had to judge the direction the route continued in (left, straight, right). For route ordering, six images taken from the study phase movie had to be placed in the correct order. For the distance comparison test, participants had to judge the absolute distance between two pairs of images taken from the route and decide which distance is the shortest. For scene recognition, route continuation and distance comparison trials of the VT test, the number of correct answers was registered. For the route ordering trial, a temporal order score was calculated, where points were awarded when the temporal location of an item was higher than the direct antecedent of the item in the reconstructed order (*n* − 1). In total, the OLMT and VT test took 20 min to complete. In Figure 2, examples of the VT tasks can be seen.

### 2.3. Standard Neuropsychological Test Battery

Global cognitive functioning was screened by means of the Mini-Mental State Examination [48]. Experienced neuropsychologists administered neuropsychological tests within six cognitive domains: memory (Rey Auditory Verbal Learning Test (RAVLT)—Dutch version [49] and Visual Association Test (VAT) [50]), attention and mental processing speed (Trail Making Test (TMT)-A [51], Letter Digit Substitution Test (LDST) [52] and Digit Span forwards [53]), executive functioning (Digit Span backwards, TMT-B and phonological letter fluency [54]), language (VAT naming, categorical animal fluency [55]), visuospatial functioning (Number Location trial of the Visual Object and Space Perception (VOSP) [56] and apraxia (ideational and ideomotor)).

### 2.4. Data Analysis

Statistical analyses were performed using SPSS Statistics 24.0 (IBM Corp., Armonk, NY, USA). We set the significance level at *p* < 0.05 (2-tailed) across all comparisons, and used a Bonferroni correction for multiple comparisons. We compared demographic data between groups by means of one-way analyses. We analyzed differences in sex between groups using Pearson χ^2^ tests. For ease of interpretation, we normalized all raw test scores to the SCD group (i.e., individual test score minus the mean of the SCD group, divided by the SD of the SCD group). For the correlational analyses, we calculated domain scores based on the individual z-scores within that domain (see section “Standard neuropsychological test battery”). Due to non-normality of the data, a log transformation was applied to the raw test scores on the OLMT test. Four univariate analyses were performed with the log absolute error and log best-fit scores of the memory and perception trials as dependent variables, and group (i.e., AD, MCI, SCD) as an independent variable, with age as a covariate. The univariate analysis on the memory trial was repeated with the log absolute error of the perception trial as additional covariate. This was to check whether differences between groups and OLMT aspects on the memory trial still existed after controlling for spatial perceptual abilities.

Due to non-normality of the data, even after transformation, the z-scores’ VT trials were analyzed with four nonparametric Kruskal–Wallis tests with group as the independent variable and scene recognition, route continuation, temporal order score and distance comparison scores as dependent variables. Post-hoc nonparametric Mann–Whitney pairwise comparisons were performed with a Bonferroni correction.

Spearman correlation coefficients were calculated for association between the six cognitive domains and the experimental OLMT and VT trials. For the OLMT test, two composite scores were calculated for the perception and memory trial, based on the mean of the absolute error rates and best-fit scores.

## 3. Results

### 3.1. Demographics

Demographic data for patients with mild AD dementia, MCI and SCD are shown in Table 1. Patients with MCI were older than participants with SCD (*p* = 0.01). Nine out of twenty-one patients with mild AD dementia were younger than 65. All groups differed on MMSE and CDR (*p* < 0.05). There were no differences in educational level.

### 3.2. Object Location Memory

#### 3.2.1. Perception Trial

Table 2 shows mean z-scores on the OLMT perception and memory trial (see Appendix A for raw data). The SCD group has been left out as they had means of zero and standard deviations of 1 by definition. There was a significant effect of group on the log mean absolute error rates (F(2, 58) = 5.0, *p* = 0.01, η^2^ = 0.15) and log mean best-fit scores of the perception trial (F(2, 58) = 4.5, *p* = 0.02, η^2^ = 0.13) (Figure 1). Post-hoc Bonferroni tests revealed that patients with AD dementia had higher mean absolute error rates and best-fit scores than participants with SCD (*p* < 0.05). There was no difference between patients with MCI and mild AD dementia or between patients with MCI and participants with SCD.

#### 3.2.2. Memory Trial

There was a significant main effect of group on the log mean absolute error rates (F(2, 55) = 33.54, *p* < 0.01, η^2^ = 0.55) and log mean best-fit scores of the memory trial (F(2,55) = 15.02, *p* < 0.01, η^2^ = 0.36) (Figure 3). Post-hoc Bonferroni tests revealed that patients with AD dementia and MCI had higher mean absolute error rates and best-fit scores than participants with SCD (all *p* < 0.01) (see Appendix A for raw data). There were no differences between patients with MCI and mild AD dementia (*p* = 0.56). After adding the mean absolute error score or best-fit scores on the perception trial as a covariate to the analysis, the effect of group remained significant on the log mean absolute error rates (F(2, 54) = 28.70, *p* < 0.01) and mean best-fit scores (F(2, 54) = 12.60, *p* < 0.01) of the memory trial. Post-hoc Bonferroni tests revealed that patients with mild AD dementia and MCI had higher mean absolute error rates and best-fit scores than controls (all *p* < 0.01), but there were no differences between patients with mild AD dementia and MCI (*p* = 0.81).

### 3.3. Virtual Tübingen Test

There was a difference between groups in the number of correct items on the scene recognition task (H(2) = 16.30, *p* < 0.01), the route continuation task (H(2) = 16.80, *p* < 0.01), the route ordering task (H(2) = 6.51, *p* = 0.04) and the distance comparison task (H(2) = 8.47, *p* = 0.01) (Figure 3) (see Appendix A for raw data). Post-hoc Mann–Whitney tests revealed that patients with mild AD dementia and MCI performed worse than participants with SCD on the scene recognition and route continuation tasks (*p* < 0.01). Patients with MCI performed worse on the route ordering task (*p* = 0.03). Patients with mild AD dementia showed a trend towards worse performance on the distance comparison task (*p* = 0.03), but this did not survive multiple comparisons correction. There were no differences on any of the tasks between patients with MCI and AD dementia.

### 3.4. Correlational Analyses

Table 3 shows the correlation coefficients between six neuropsychological domains and the OLMT and VT tasks. The route continuation test had a weak to moderate association with the memory domain score (r(64) = 0.40, *p* < 0.01). The temporal route ordering task had a weak to moderate association with attention (r(64) = 0.33, *p* < 0.05). The other VT tasks and the perception and memory trials of the OLMT had no significant correlation with any of the cognitive domains.

## 4. Discussion

The aim of the present study was to evaluate the differential ability of a short digital spatial memory test in patients with mild AD dementia, MCI and SCD. Our main findings are that both patients with mild AD dementia and MCI were impaired on object location memory and spatial navigation. In addition, there was little to no relation between the OLMT and VT tests and other cognitive domains, suggesting that performance on these tests reflects a different cognitive construct than is currently used in the standard diagnostic work-up. These results indicate that the short digital spatial memory test battery can detect object location memory and navigation impairment in patients with mild AD dementia and MCI, and therefore could be a valuable addition to the standard neuropsychological protocol in memory clinics.

Object location memory was equally impaired in patients with AD dementia and MCI compared to the SCD group. These findings expand on previous results showing that object location memory can be used to discriminate patients with AD dementia and MCI from healthy older individuals [43,58]. Lesion studies have shown that the hippocampus and parietal cortex play an important role in the binding of objects to locations in memory [59,60,61,62,63,64,65,66,67,68], and atrophy in these areas has consistently been found in both patients with MCI and AD dementia [13]. Notably, patients with AD dementia also showed impairment on the OLMT perception task. Visuospatial impairment has been reported previously in patients with AD dementia; however, this is seen more often in patients with an early-onset (before the age of 65) phenotype [69]. As ~40% of patients had an early-onset phenotype of AD dementia, this might have influenced our results. However, after adding the perception trial as a covariate to the analysis of the memory trial, patients with AD dementia and MCI still performed significantly worse on the memory task compared to the SCD group, suggesting that impairment stems from pure memory deficits. 

Scene recognition was impaired in both patients with mild AD dementia and MCI, compared to the SCD group. This is unsurprising, as scene recognition has been previously associated with the parahippocampal cortex [59,68,70,71,72], an area that has been shown to deteriorate in the early stages of AD dementia, but also MCI [73,74]. Similarly, route continuation was also impaired in both patients with mild AD dementia and MCI compared to participants with SCD. Route continuation in the current study is a measure for route learning/egocentric navigation, and impairment on such tasks in AD dementia and MCI has been associated with a parieto-frontal network, including the frontal gyrus, parietal sulcus and the anterior cingulate cortex [75,76]. Indeed, previous studies have shown severe impairments in scene recognition, remembering turns at decision points and recalling the temporal order of routes in both patients with AD dementia and MCI. Furthermore, some of these studies have found that route continuation and route ordering are impaired in both patients with AD dementia and MCI, to the same extent [11,14,24,27,29,33,42,61,77].

Surprisingly, we found impairment on temporal route ordering in patients with MCI, but not patients with mild AD dementia. Likewise, patients with MCI were not impaired on the distance comparison task, and in patients with AD dementia, a difference compared to SCD did not survive correction for multiple comparisons. Both temporal order memory, from an egocentric reference frame, and spatial navigation tasks, from an allocentric reference frame, such as the distance comparison task, have been associated with the hippocampus [78,79] and parietal areas in patients with MCI and AD dementia [80]. Patients in the symptomatic stage should be especially impaired on tasks that are dependent on these areas, such as temporal order memory and distance comparison. Other studies have shown specific impairment in allocentric navigation already at the MCI stage [41]. Possible reasons for why we did not find the expected differences are small sample sizes, a small number of test items in the route ordering and distance comparison task (i.e., six and three, respectively) or a difference in the strategies used by patients. It could be that temporal order memory was impaired to such an extent in patients with AD dementia that patients relied, for example, on a primacy/recency effect strategy, where only items that were seen earliest/latest in the route were remembered correctly. As there was a fixed starting position (i.e., all patients were presented the images in the same order), and several patients with AD dementia claimed to remember nothing and left the images as they were initially presented, this possibly resulted in them having a higher temporal route ordering score than patients who did attempt to re-order the images. Yet, there was no significant difference between patients with AD dementia and MCI (who were impaired on temporal route ordering) suggesting that there is a trend towards impairment on temporal order memory in patients with mild AD dementia.

Another explanation for why we did not find impairment on the distance comparison task in patients with MCI could be that we did not differentiate between amnestic, non-amnestic and multi-domain MCI. Previous studies investigating the distinction between egocentric and allocentric spatial navigation in MCI differentiated between subtypes of MCI [26,28,38]. For example, Hort and colleagues (2007) divided patients with MCI in an amnestic type, a multi-domain type and a non-amnestic type. The results showed that patients with amnestic MCI were especially impaired on the allocentric test compared to the patients with non-amnestic MCI [26]. This is unsurprising, given that patients with amnestic MCI are identified as those most likely to convert to symptomatic AD, whereas most other forms remain stable, revert to a normal stage or convert to another form of dementia [81]. Patients with MCI included in the current study were not stratified according to phenotype during the diagnosis setting, and so the inclusion of all three types could have influenced our results. However, as patients with MCI were significantly impaired on the hippocampus-dependent route ordering task, indicating that patients with amnestic MCI were representative of the sample, it seems more likely that the absence of impairment on the distance comparison task was due to the low number of test items, and a task with more items might be more sensitive.

There were no correlations found between the OLMT tests and any of the measured cognitive domains that are part of standard neuropsychological clinical assessment. There was a weak to moderate correlation between the route continuation task and memory, and interestingly, also between the temporal route ordering task and attention scores. A possible explanation for the latter is that the temporal route ordering task also depends on frontal brain areas, such as the prefrontal cortex. Several previous studies have demonstrated the involvement of prefrontal areas in spatial navigation planning [82,83,84], and these areas also play a critical role in attentional processes [85]. The fact that only very few weak to moderate correlations with other neuropsychological domains were found suggests that the OLMT and VT tests measure cognitive constructs that are not well represented within routine neuropsychological assessment. This indicates the importance of introducing spatial memory tests in clinical practice.

There are several limitations to this study, including small sample sizes, not-age-matched groups and the lack of neuroimaging and CSF biomarker information. Patients with MCI have often been shown to possess neuropathological signs of AD and progress to AD dementia at a faster rate than cognitively healthy individuals [86]. MCI patients with underlying AD pathology exhibit the amnestic phenotype more often than patients with non-amnestic MCI [87], but we included all patients with MCI regardless of phenotype. Similarly, participants with SCD had cognitive complaints, which motivated their consultation at the memory clinic, thereby meeting the SCD plus criteria and having a higher risk of having preclinical AD [86]. Although both MCI and SCD are heterogeneous syndromes, and not all individuals will develop AD dementia, using participants with SCD as a control group and the non-specificity of MCI phenotypes (i.e., amnestic, non-amnestic and multi-domain) are drawbacks in this study. In addition, our speculations about the potential neural mechanisms that underlie the differences between patient groups cannot be corroborated in the absence of neuroimaging data. Another limitation of this study is that participants with SCD were significantly younger than patients with MCI, including two participants younger than 50. Age-related decline has been found in spatial memory and navigation studies in healthy older individuals [88,89], and including young participants in the SCD group may have affected the results. Group differences should therefore be interpreted cautiously, especially in the VT test, as those were not corrected for age due to the nonparametric statistical approach. Future research should focus on validating these tests in a larger cohort of patients, taking into account biomarker information on underlying AD pathology, correlating with brain imaging data and comparing performance with a population of cognitively healthy older adults without underlying AD pathology. In addition, longitudinal follow-up is necessary to investigate the potential of the OLMT and VT test as cognitive markers in AD development. The development of short, computer-generated and yet sensitive spatial memory tests is important if we are to introduce tests in clinical practice that reflect real-world performance, but at the same time are easy to administer.

In conclusion, object location memory and spatial navigation were found to be clearly affected in patients with MCI and mild AD dementia. Neither object location memory nor navigation assessments are generally performed in current clinical practice. Given the importance of these functions and the frequency of deficits, evaluation of spatial memory abilities should be included in standard neuropsychological examination in older adults. Yet, at present, no standardized spatial memory measures are in use. The Object Relocation program and virtual reality tests offer a viable opportunity for this purpose. This study confirms that object location memory and spatial navigation are impaired in mild AD dementia and MCI, and underlines that assessment of these processes should be incorporated in the standard diagnostic work-up for older adults.

## Figures and Tables

**Figure 1 brainsci-11-01350-f001:**
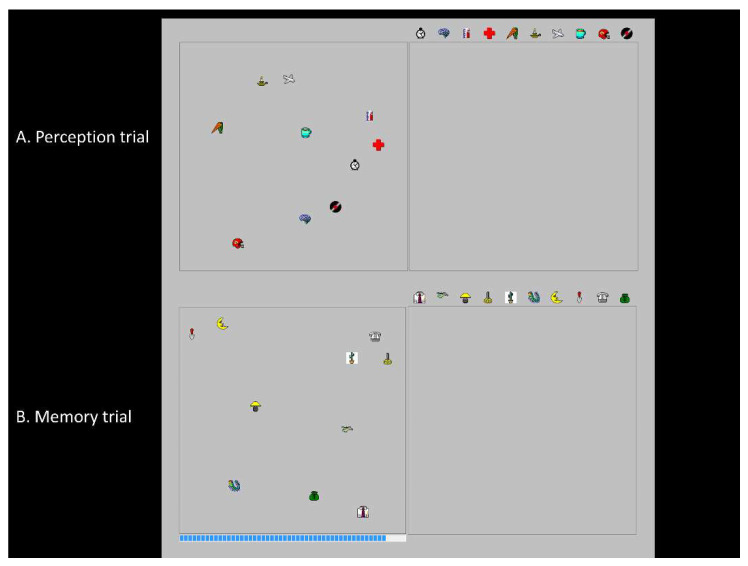
(**A**) the perception trial, and (**B**) the memory trial of the OLMT.

**Figure 2 brainsci-11-01350-f002:**
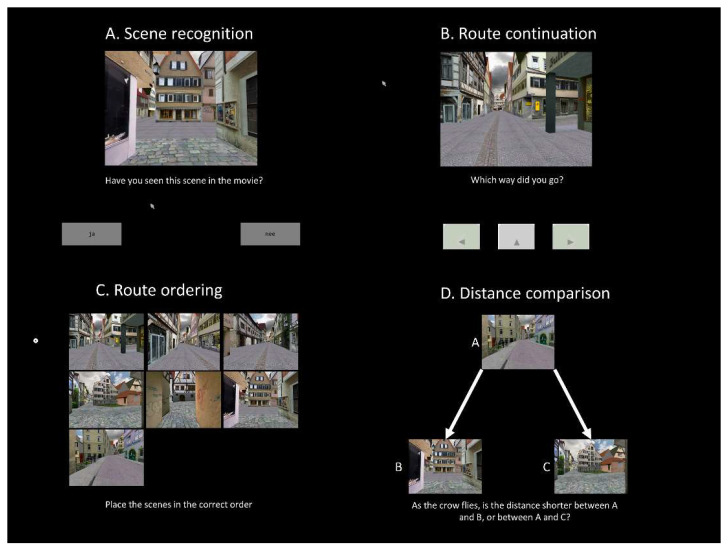
Examples of (**A**) the scene recognition, (**B**) the route continuation, (**C**) the route ordering and (**D**) the distance comparison task.

**Figure 3 brainsci-11-01350-f003:**
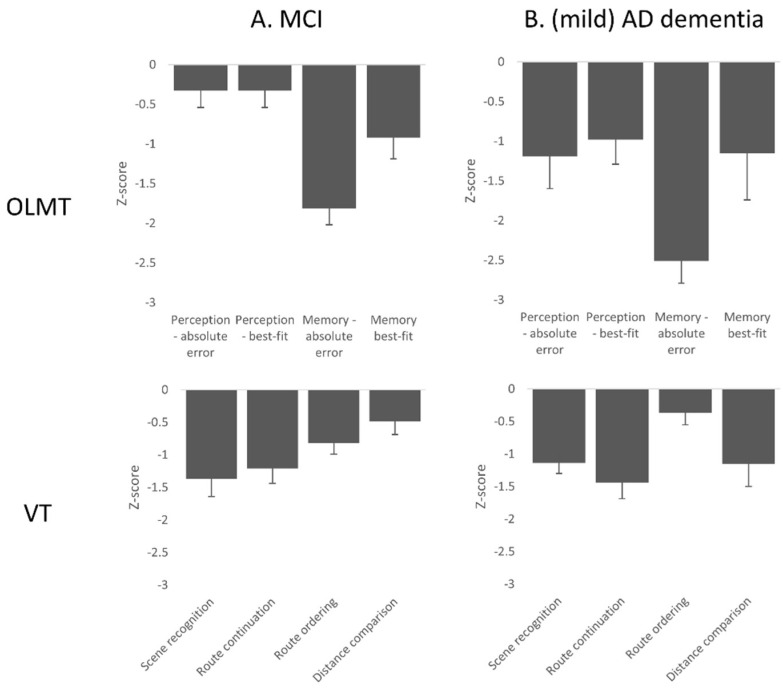
Bar plot of the means and standard errors on the OLMT and VT tasks for (**A**) patients with MCI, and (**B**) patients with mild AD dementia. Abbreviations: MCI = Mild Cognitive Impairment; AD = Alzheimer’s Disease.

**Table 1 brainsci-11-01350-t001:** Demographic data (mean + standard deviation).

	SCD (*n* = 21)	MCI (*n* = 22)	(Mild) AD Dementia (*n* = 21)
Age [Age range]	62.2 ± 8.9 ^b^ (39–75)	69.6 ± 8.3 ^a^ (52–81)	66.9 ± 6.9 (57–82)
Women, *n* (%)	10 (47.6%)	10 (45.5%)	10 (47.6%)
Educational level *	5.7 ± 1.2	5.1 ± 1.4	5.3 ± 1.3
Mini-Mental State Examination	29.1 ± 1.1 ^bc^	26.1 ± 2.8 ^ac^	22.8 ± 5.1 ^ab^
Clinical Dementia Rating scale	0.1 ± 0.2 ^bc^	0.50 ± 0.00 ^ac^	0.9 ± 0.2 ^ab^

Abbreviations: SCD = Subjective Cognitive Decline; MCI = Mild Cognitive Impairment; AD = Alzheimer’s disease; ^a^ Significant difference at *p* < 0.05 with SCD; ^b^ significant difference at *p* < 0.05 with MCI; ^c^ significant difference at *p* < 0.05 with AD. * Dutch educational system categorized into levels from 1 = 5 less than 6 years of primary education to 7 = 5 academic schooling [57].

**Table 2 brainsci-11-01350-t002:** Mean z-scores and standard deviations on the OLMT and VT tests.

	*n*	MCI	*n*	(Mild) AD Dementia
Object location memory				
Perception—absolute error	22	−0.33 ± 0.99	19	−1.19 ± 1.80 **^a^
Perception—best fit	22	−0.33 ± 0.99	19	−0.98 ± −1.36 **^a^
Memory—absolute error	20	−1.81 ± 0.92 **^a^	17	−2.51 ± 1.15 **^a^
Memory—best fit	20	−1.47 ± 1.19 **^a^	17	−2.68 ± 2.45 **^a^
Virtual Tübingen				
Scene recognition	22	−1.37 ± 1.25 **^a^	21	−1.14 ± 0.73 **^a^
Route continuation	22	−1.21 ± 1.07 **^a^	21	−1.44 ± 1.13 **^a^
Route ordering	21	−0.82 ± 0.78 **^a^	20	−0.37 ± 0.82
Distance comparison	21	−0.49 ± 0.90	19	−1.15 ± 1.53 *

Abbreviations: MCI = Mild Cognitive Impairment; AD = Alzheimer’s disease. * *p* < 0.05, ** *p* < 0.01. ^a^ Significant difference with subjective cognitive decline group.

**Table 3 brainsci-11-01350-t003:** Spearman correlation coefficients between composite neuropsychological domain scores and the OLMT and VT test scores.

	Memory	Attention	Executive	Language	Visuospatial	Apraxia
Object location memory						
Perception	0.25	0.15	0.23	0.11	0.06	0.09
Memory	0.02	0.13	0.00	−0.04	0.23	−0.06
Virtual Tübingen						
Scene recognition	0.26	0.12	0.19	0.11	0.04	0.03
Route continuation	0.40 **	0.21	0.18	0.19	0.08	−0.00
Temporal route ordering	0.21	0.33 *	0.15	0.26	0.15	0.20
Distance comparison	−0.28	0.11	0.16	0.12	0.03	0.30

* *p* < 0.05, ** *p* < 0.01.

## Data Availability

Data will be shared upon request.

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
