# Peer review of "Short Digital Spatial Memory Test Detects Impairment in Alzheimer’s Disease and Mild Cognitive Impairment"

_brainsci, 2021, doi:10.3390/brainsci11101350_

Round 1

Reviewer 1 Report

The authors used a battery of short digital spatial memory tests to detect impairment in AD and MCI patients compared to the SCD patients. Given that spatial cognition is impaired very early in AD and seems to be a different construct compared to the traditionally used neuropsychological tests, it is very important to find a easy-to-do and sensitive spatial memory/navigation test to diagnose AD in the earliest stages. This manuscript fits perfectly into the topic of this special issue. However, there are several serious issues that should be addressed. They are listed in the attached document.

Major comments

Abstract

1) It is not clear if scene recognition, route continuation, etc. relate to the VT test.

Introduction

1) AD as a disease and clinical staging of AD are used interchangingly and should be more precisely clarified throughout the manuscript. AD as a clinical pathological entity encompasses a preclinical (presymptomatic) stage (described as preclinical AD) and symptomatic stages. The clinical stages include a MCI stage (MCI due to AD; other term for this is prodromal AD) and a dementia stage (AD dementia) that is based on the severity of impairment classified as mild, moderate and severe dementia.

Materials and Methods

1) Inclusion and exclusion criteria for participants should be described in more detail. It should be clarified that patients with probable AD were in the dementia stage. The criteria that were used for establishing MCI and AD (dementia?) diagnosis in this study should be specified as cited Albert's and McKhann's criteria do not define neuropsychological tests and cut-offs for establishing the diagnosis. Did the study include all MCI patients, amnestic and non-amnestic, and what was their proportion? It should be specified if patients with other diagnoses as Parkinson's disease, stroke, brain trauma were excluded and if patients with vascular impairment (based on MRI brain scan or Hatchinsky scale) and depression were included or not. Also it should be specified what means "normal" clinical investigations for SCD (MRI, laboratory, neuropsychology - compared to normative data?).

2) Using SCD patients as a control group is a drawback of the study as patients with cognitive complaints, which motivate the consultation at the clinic, mostly meet the SCD plus criteria and so are at higher risk of having preclinical AD. This should be stated as a limitation of the study and throughout the manuscript it should be clearly stated the spatial tests in the current study differentiated SCD patients, who potentially have preclinical AD from patients with MCI and AD dementia but it is not possible to infer from these results how these tests can distinguish healthy older adults from the these patients. It should be clearly stated throughout the manuscript (including tables) that MCI and AD patients were compared to the SCD patients

3) One of the exclusion criteria was CDR >= 1, however the mean and SD of CDR in AD patients was 0.9 and 0.2, which means that most of AD patients had CDR 1.

4) Bonferroni correction should be used in the correlational analysis and results surviving this correction should be reported.

Results

1) Results of the post-hoc tests in the memory trial of the OLMT task after correction for the perception trial should be reported to see if the differences, especially between the MCI and SCD groups remained significant. This should be also discussed in the discussion.

2) Significant p values for post-hoc tests in the VT task should be reported.

3) In the perception and memory trials of the OLMT the authors state that AD patients and patients with MCI and AD, respectively, reversed relative locations of objects. This is an interpretation that should be moved to the discussion. In addition, the z-scores of absolute and best fit errors are very close to each other in each group, especially in the perception trial of the AD group (I suppose that they would not statistically differ from each other) and thus do not support this statement.

4) A range of correlation coefficients should be reported together with p values in the correlational analyses.

Discussion

The discussion should be modified according to the comments below:

1) The statements in the first paragraph are not supported by data and design of the study - the authors did not evaluate the AD spectrum but patients with AD dementia and patients with SCD and MCI, whicht may be caused by various etiology. The absence of differences between AD and MCI patients do not infer that object location memory and spatial navigation declines progressively at the MCI stage and previous studies found differences between MCI and AD dementia patients. There were some moderate correlations with cognitive functions. If these correlations would not survive corrections for multiple comparisons, the statement about the different cognitive construct would be more sound. The authors did not perform ROC analysis to support the statement that this battery is sensitive to detect impairment in AD and MCI aptients. 

2) In the second paragraph, similar object location memory in AD and MCI patients is not in line with findings of gradually increasing hippocampal atrophy from MCI to AD. As mentioned in the previous comment, I am not sure that the data support that AD patients switched objects in the perception task. Also I am not sure why early onset AD is mentioned here as the mean age of AD group was 67 years. The post-hoc results of the OLMT memory task with the perception trial as a covariate are not reported so, it is not possible to find out if the statement that AD and MCI patients performed worse than the SCD group that is stemming from pure memory deficits is true or not.

3) In the third paragraph, scene recognition, route continuation and temporal route ordering should be discussed separately as they form different constructs that are supported from separate brain regions - scene (landmark) recognition is supported by parahippocampal cortex, route continuation in this task is a measure for route-learning/egocentric navigation that is supported by posterior parietal cortex and caudate nucleus and temporal order memory strongly depends on the hippocampus (Iglóy 2010, PNAS). It is very surprising that AD patients, where hippocampal atrophy and dysfunction is a hallmark of the disease, were not impaired in a hippocampus-dependent task (temporal order memory). This has to be thoroughly discussed.

4) To support the statement in the fourth paragraph that similar performance in allocentric distance comparison task in MCI and SCD patients is caused by heterogeneity of the MCI group, the authors should report in the Methods section the proportions of each MCI subtype. If the study recruited more non-amnestic than amnestic MCI patients, this argumentation would be very convincing otherwise it would be appropriate to provide further explanation. We have to also keep in mind that MCI patients were impaired in temporal route ordering that is supported by the hippocampus, which also strongly supports allocentric navigation. If the explanation above was true, MCI patients should not be impaired in this task.

5) To strengthen the statement in the fourth paragraph of a different construct of the VT task, a correction for multiple comparisons should be applied. It is also of interest why hippocampus-dependent temporal route ordering unlike other tasks correlated with attention and also why there is a negative correlation (although small) between the allocentric distance comparison task and memory, both of which should be supported by the hippocampus.

6) Further limitations are the absence of information about underlying disease in MCI and SCD patients, using SCD patients as a control group, absence of brain imaging data, not evaluating depressive symptoms and including patients with vascular lesions (if applicable).

7) In the Conclusion section statement that this assessment could "could significantly help in the diagnosis of patients with AD, MCI and possibly its subtypes" is not supported by the data.

Table 2

1) According to the Results section route ordering is not impaired in AD group but here is indicated as significantly different from SCD group.

Table 3

1) Results surviving correction for multiple comparisons (Bonferroni or less conservative Holm-Bonferroni) should be indicated.

Minor comments

Abstract

1) Abbreviation AD is not explained as first appears.

2) The space before Methods is missing.

3) The range of correlation coefficients is not written in a standard format.

Materials and Methods

1) Figures with VT and OLM tasks would be helpful for better understanding the principle of the tests.

2) It is not clear how many crossing points were on the route.

Discussion

1) In the Conclusion section the abbreviation "OR" is not explained.

Table 1

1) The cross referring to the explanation of the abbreviation "MCI" should not be placed near Controls.

2) If asterisk is added to all groups, does it mean that all groups differ from each other?

Table 2

1) OLMT task is incorrectly abbreviated as ORT

2) Superscript “b” indicating that the particular group differs from the MCI group is attached to the value of the MCI group and the superscript c indicating that the particular group differs from the AD group is not present in the table

Table 3

1) OLMT task is incorrectly abbreviated as ORT

Table A1

1) SD of perception - best fit in AD group has a negative value, which is not possible.

Author Response

Major comments

Abstract

  • It is not clear if scene recognition, route continuation, etc. relate to the VT test.

Reply: we have added an explanation to the methods section of the abstract addressing this (p. 1, lines 22-24).

Introduction

  • AD as a disease and clinical staging of AD are used interchangingly and should be more precisely clarified throughout the manuscript. AD as a clinical pathological entity encompasses a preclinical (presymptomatic) stage (described as preclinical AD) and symptomatic stages. The clinical stages include a MCI stage (MCI due to AD; other term for this is prodromal AD) and a dementia stage (AD dementia) that is based on the severity of impairment classified as mild, moderate and severe dementia.

Reply: We agree with the reviewer that this should be more precisely explained in the manuscript and that we should not use those concepts interchangeably. We have added an explanation of the clinical stages of AD to the Introduction (p.2 lines 47-51), and clarified that we included patients with mild AD dementia and patients with MCI throughout the manuscript. 

Materials and Methods

  • Inclusion and exclusion criteria for participants should be described in more detail. It should be clarified that patients with probable AD were in the dementia stage. The criteria that were used for establishing MCI and AD (dementia?) diagnosis in this study should be specified as cited Albert's and McKhann's criteria do not define neuropsychological tests and cut-offs for establishing the diagnosis. Did the study include all MCI patients, amnestic and non-amnestic, and what was their proportion? It should be specified if patients with other diagnoses as Parkinson's disease, stroke, brain trauma were excluded and if patients with vascular impairment (based on MRI brain scan or Hatchinsky scale) and depression were included or not. Also it should be specified what means "normal" clinical investigations for SCD (MRI, laboratory, neuropsychology - compared to normative data?).

Reply: We have specified the requested information in section 2.1 of the Materials and Methods (p.3, lines 99-118). Clinical diagnoses were established based on all available data, including MRI, EEG and neuropsychological assessment. Although, performance on neuropsychological assessment plays a crucial role in the diagnosis setting, this is judged based on clinical expertise rather than specific cut-off scores (as described in the NIA-AA criteria). We have clarified the diagnostic process in more detail in section 2.1.

  • Using SCD patients as a control group is a drawback of the study as patients with cognitive complaints, which motivate the consultation at the clinic, mostly meet the SCD plus criteria and so are at higher risk of having preclinical AD. This should be stated as a limitation of the study and throughout the manuscript it should be clearly stated the spatial tests in the current study differentiated SCD patients, who potentially have preclinical AD from patients with MCI and AD dementia but it is not possible to infer from these results how these tests can distinguish healthy older adults from the these patients. It should be clearly stated throughout the manuscript (including tables) that MCI and AD patients were compared to the SCD patients

Reply: We agree with the reviewer that using the participants with SCD as a control group is a drawback of this study and should be properly addressed throughout the manuscript. We added a more in-depth discussion on this topic to the Discussion (p. 13, lines 472-482), and replaced ‘controls’ with ‘participants with SCD’ throughout the manuscript. In a separate study, we administered the same spatial memory paradigm to cognitively healthy older adults via the experimental psychology department of Utrecht University. As this data was collected in a different setting and study, we did not include it in the current manuscript. If preferred by the reviewer/editor, we can add this to a Supplementary file.

  • One of the exclusion criteria was CDR >= 1, however the mean and SD of CDR in AD patients was 0.9 and 0.2, which means that most of AD patients had CDR 1.

Reply: this was a typo in the manuscript. The exclusion criteria was CDR>1. We adjusted this in section 2.1 of Materials and Methods (p. 3, lines 117-118).

  • Bonferroni correction should be used in the correlational analysis and results surviving this correction should be reported.

Reply: We adjusted accordingly in section 3.4 of the Results (p. 6, lines 255-259; p. 10, Table 3)

Results

  • Results of the post-hoc tests in the memory trial of the OLMT task after correction for the perception trial should be reported to see if the differences, especially between the MCI and SCD groups remained significant. This should be also discussed in the discussion.

Reply: We have added this to section 3.2.2 of the Results (p.7, lines 232-237) and address this in the Discussion (p.11, lines 370-374).

  • Significant p values for post-hoc tests in the VT task should be reported.

Reply: we have added p values for post-hoc tests to section 3.3 of the Results (p. 7, lines 247 and 249).

  • In the perception and memory trials of the OLMT the authors state that AD patients and patients with MCI and AD, respectively, reversed relative locations of objects. This is an interpretation that should be moved to the discussion. In addition, the z-scores of absolute and best fit errors are very close to each other in each group, especially in the perception trial of the AD group (I suppose that they would not statistically differ from each other) and thus do not support this statement.

Reply: We agree with the reviewer that this cannot be inferred without statistical tests. A pairwise sample t-test comparing the absolute mean errors and best-fit score within each group, however, is not possible as the scores are derived from the same trials. Therefore, we deleted the interpretation from sections 2.2 (p. 4, lines 139-141), 3.2.1 (p. 6, lines 216-221) and 3.2.1. (p. 7, lines 229-231).

  • A range of correlation coefficients should be reported together with p values in the correlational analyses.

Reply: We have added the correlation coefficients to section 3.4 (p. 7, lines 257 and 259).

Discussion

The discussion should be modified according to the comments below:

  • The statements in the first paragraph are not supported by data and design of the study - the authors did not evaluate the AD spectrum but patients with AD dementia and patients with SCD and MCI, whicht may be caused by various etiology. The absence of differences between AD and MCI patients do not infer that object location memory and spatial navigation declines progressively at the MCI stage and previous studies found differences between MCI and AD dementia patients. There were some moderate correlations with cognitive functions. If these correlations would not survive corrections for multiple comparisons, the statement about the different cognitive construct would be more sound. The authors did not perform ROC analysis to support the statement that this battery is sensitive to detect impairment in AD and MCI aptients. 

Reply: We agree with the reviewer and rewrote the first paragraph (p. 11, lines 346-347, 348-350, 354, 355). We applied a Bonferroni correction on the correlational analyses, and found that most correlations did not survive multiple testing. We agree with the reviewer that this makes the statement that performance on the OLMT and VT tasks reflects different cognitive constructs more sound.

  • In the second paragraph, similar object location memory in AD and MCI patients is not in line with findings of gradually increasing hippocampal atrophy from MCI to AD. As mentioned in the previous comment, I am not sure that the data support that AD patients switched objects in the perception task. Also I am not sure why early onset AD is mentioned here as the mean age of AD group was 67 years. The post-hoc results of the OLMT memory task with the perception trial as a covariate are not reported so, it is not possible to find out if the statement that AD and MCI patients performed worse than the SCD group that is stemming from pure memory deficits is true or not.

Reply: we have tempered and rewritten our conclusions in the second paragraph. We removed the statement about switching objects in the perception task (p. 11, lines 366-367). We added age range to Table 1 (p. 7), added the number of patients with AD dementia that were younger than 65 to section 3.1 of the Results (p. 6, lines 203-204), and elaborate on this in the Discussion (p. 11, lines 369-370). We report the post-hoc results of the OLMT memory task with the perception trial as covariate to section 3.2.2 (p. 7, lines 233-237), which corroborates our statement about impairment on the memory trial stemming from pure memory deficits (p. 11, lines 370-374).

  • In the third paragraph, scene recognition, route continuation and temporal route ordering should be discussed separately as they form different constructs that are supported from separate brain regions - scene (landmark) recognition is supported by parahippocampal cortex, route continuation in this task is a measure for route-learning/egocentric navigation that is supported by posterior parietal cortex and caudate nucleus and temporal order memory strongly depends on the hippocampus (Iglóy 2010, PNAS). It is very surprising that AD patients, where hippocampal atrophy and dysfunction is a hallmark of the disease, were not impaired in a hippocampus-dependent task (temporal order memory). This has to be thoroughly discussed.

Reply: We have rewritten the third paragraph of the Discussion (p. 12, lines 393-416). We have discussed each VT task separately and relate this to the underlying neural mechanisms. We thank the reviewer for the suggested reference, and included it in the manuscript. We also provide a more detailed discussion of why we did not find the expected differences between patients with AD dementia and SCD on the route ordering task.

  • To support the statement in the fourth paragraph that similar performance in allocentric distance comparison task in MCI and SCD patients is caused by heterogeneity of the MCI group, the authors should report in the Methods section the proportions of each MCI subtype. If the study recruited more non-amnestic than amnestic MCI patients, this argumentation would be very convincing otherwise it would be appropriate to provide further explanation. We have to also keep in mind that MCI patients were impaired in temporal route ordering that is supported by the hippocampus, which also strongly supports allocentric navigation. If the explanation above was true, MCI patients should not be impaired in this task.

Reply: Unfortunately, patients with MCI were not stratified according to specific phenotype during the diagnosis setting. It would be possible to determine the specific phenotype using the neuropsychological data, but this would be in comparison to the SCD group rather than normative data. Therefore, we decided to not retrospectively determine specific MCI phenotypes. However, we have rewritten the paragraph covering this topic in the Discussion (p. 12, lines 417-433). We agree with the reviewer that the impairment on the route ordering task in patients with MCI is an indication of hippocampal dysfunction and thus the amnestic phenotype, and have added this to the paragraph (p. 12, lines 429-433). Lastly, we discuss the lack of information on the proportion of each MCI phenotype as a study limitation more thoroughly (p.13 , lines 472-484).

  • To strengthen the statement in the fourth paragraph of a different construct of the VT task, a correction for multiple comparisons should be applied. It is also of interest why hippocampus-dependent temporal route ordering unlike other tasks correlated with attention and also why there is a negative correlation (although small) between the allocentric distance comparison task and memory, both of which should be supported by the hippocampus.

Reply: We corrected for multiple comparisons in the correlational analyses (p. 7, lines 255-260). The correlation between the distance comparison task and memory was no longer significant. We added an explanation for the significant correlation between temporal route ordering and attention (p. 13, lines 462-466).

  • Further limitations are the absence of information about underlying disease in MCI and SCD patients, using SCD patients as a control group, absence of brain imaging data, not evaluating depressive symptoms and including patients with vascular lesions (if applicable).

Reply: We added the absence biomarker information, such as brain imaging data, as well as the use of participants with SCD as control group as limitations to the Discussion (p. 13, lines 471-484). Furthermore, we added a statement to section 2.1 of the Methods that we excluded patients with other neurological/psychiatric diseases, major vascular damage and traumatic brain injury (p. 3, lines 110-11).

  • In the Conclusion section statement that this assessment could "could significantly help in the diagnosis of patients with AD, MCI and possibly its subtypes" is not supported by the data.

Reply: We have rewritten the conclusion statement (p. 14, lines 505-508).

Table 2

  • According to the Results section route ordering is not impaired in AD group but here is indicated as significantly different from SCD group.

Reply: We adjusted this in Table 2 (p. 9)

Table 3

  • Results surviving correction for multiple comparisons (Bonferroni or less conservative Holm-Bonferroni) should be indicated.

Reply: We have adjusted accordingly in Table 3 (p.10)

Minor comments

Abstract

  • Abbreviation AD is not explained as first appears.

Reply: We adjusted accordingly (p.1, line 17)

  • The space before Methods is missing.

Reply: We adjusted accordingly (p.1, line 19)

  • The range of correlation coefficients is not written in a standard format.

             Reply: we adjusted accordingly (p. 1, line 32).

Materials and Methods

  • Figures with VT and OLM tasks would be helpful for better understanding the principle of the tests.

Reply: We agree with the reviewer that figures of the different tasks are helpful in understanding the set-up of the tasks. We have added two figures with illustrations of the OLMT (Figure 1; p. 4) and the VT tasks (Figure 2; p.5).

  • It is not clear how many crossing points were on the route.

Reply: we have added this to section 2.2 of the Methods (p. 4, line 149).

Discussion

  • In the Conclusion section the abbreviation "OR" is not explained.

Reply: we have specified the abbreviation (p. 14, line 504).

Table 1

  • The cross referring to the explanation of the abbreviation "MCI" should not be placed near Controls.

Reply: we have adjusted the lay-out of the table by removing the cross referring, and adding the abbreviations in a legend under the table (p. 8).

  • If asterisk is added to all groups, does it mean that all groups differ from each other?

Reply: we have clarified significant differences in Table 1 (p. 8)

Table 2

  • OLMT task is incorrectly abbreviated as ORT

Reply: we adjusted accordingly (p. 9).

  • Superscript “b” indicating that the particular group differs from the MCI group is attached to the value of the MCI group and the superscript c indicating that the particular group differs from the AD group is not present in the table

Reply: we adjusted accordingly (p. 9).

Table 3

  • OLMT task is incorrectly abbreviated as ORT

Reply: we adjusted accordingly (p. 10).

Table A1

  • SD of perception - best fit in AD group has a negative value, which is not possible.

Reply: this was a typo, that we now corrected (p. 15).

Reviewer 2 Report

Poos et al examines spatial computation and possible use of a digital battery testing such cognitive function in patients with Alzheimer’s Disease (AD) and Mild Cognitive Impairment (MCI). The question tackled here is interesting and has important clinical implications for understanding and developing effective clinical treatment of AD, MCI, and conditions with memory and learning deficits. I also appreciate the clarity with which the authors set up the questions of interest and introduce relevant background literature in the introduction. Having said this, I found certain information regarding the statistical analyses and results to be missing. Moreover, I also believe that a few additional speculations about what the observed findings may tell us about the disrupted neural mechanisms in AD, MCI, as well as other related conditions could significantly strengthen the conclusions and interpretation of the reported findings.

Comments:

  1. Out of curiosity, due to the non-normality of the patient data as noted by the authors in Data analysis section, do the authors believe that the results would have been different if a non-parametric statistical test were to be performed? And why?
  2. Could the authors please consider making a figure of the main findings (e.g., bar plot, violin plot)? While all results are reported in the tables, it is difficult for the size/strength of differences between conditions to be directly visualized and compared.
  3. While the present study focuses on behavioral measures of spatial cogition, it would be extremely helpful to readers if the authors could offer some speculation about the potential neural mechanisms that underlie the differences observed between younger and older adults. Such formulation/hypothesis would also benefit future studies that wish to examine spatial navigation and other related cognitive functions in healthy and clinical populations. It will also complement the note on limitations of the study nicely. For example, various aspects of spatial cognition have been investigated using fMRI, MEG, as well as in hippocampus-lesioned patients to examine the extent to which such functions are subserved by structures within the medial temporal lobes. Below are some sample references that complement the current behavioral findings and could lay foundation for how the current findings could relate to the underlying neural dynamics.
  • Emily Ruzich, Crespo-Garcia, Sarang S Dalal, Justin F Schneiderman (2019). Human Brain Mapping. Characterizing hippocampal dynamics with MEG: A systematic review and evidence‐based guidelines
  • Nuttida Rungratsameetaweemana & Larry Squire (2018). Learning & Memory. Preserved capacity for scene construction and shifts in perspective after hippocampal lesions.
  • Tom Hartley, Colin Lever, Neil Burgess, John O’Keefe (2014). Philos Trans R Soc Lond B Biol Sci. Space in the brain: how the hippocampal formation supports spatial cognition.

Author Response

Poos et al examines spatial computation and possible use of a digital battery testing such cognitive function in patients with Alzheimer’s Disease (AD) and Mild Cognitive Impairment (MCI). The question tackled here is interesting and has important clinical implications for understanding and developing effective clinical treatment of AD, MCI, and conditions with memory and learning deficits. I also appreciate the clarity with which the authors set up the questions of interest and introduce relevant background literature in the introduction. Having said this, I found certain information regarding the statistical analyses and results to be missing. Moreover, I also believe that a few additional speculations about what the observed findings may tell us about the disrupted neural mechanisms in AD, MCI, as well as other related conditions could significantly strengthen the conclusions and interpretation of the reported findings.

Comments:

  1. Out of curiosity, due to the non-normality of the patient data as noted by the authors in Data analysis section, do the authors believe that the results would have been different if a non-parametric statistical test were to be performed? And why?

Reply: We performed and report non-parametric statistical tests on the VT tests and non-parametric correlational analyses for the association with other cognitive domains. We have now clarified this in section 2.4 of the Methods (p. 6, lines 192 and 194). To adequately deal with non-normality of the OLMT data, we performed a log transformation on the absolute error and best-fit scores. We did not expect to find different results if we would have used non-parametric statistical tests on the OLMT data, but then we could not add the perception trial as a covariate to demonstrate that deficits on the memory trial are stemming from pure memory deficits.

  1. Could the authors please consider making a figure of the main findings (e.g., bar plot, violin plot)? While all results are reported in the tables, it is difficult for the size/strength of differences between conditions to be directly visualized and compared.

Reply: We agree with the reviewer that a figure of the main findings helps in clarifying the size of the differences between conditions. Therefore, we added a bar plot with the means and standard errors on the OLMT and VT tasks to the manuscript (Figure 3; p. 8).

  1. While the present study focuses on behavioral measures of spatial cogition, it would be extremely helpful to readers if the authors could offer some speculation about the potential neural mechanisms that underlie the differences observed between younger and older adults. Such formulation/hypothesis would also benefit future studies that wish to examine spatial navigation and other related cognitive functions in healthy and clinical populations. It will also complement the note on limitations of the study nicely. For example, various aspects of spatial cognition have been investigated using fMRI, MEG, as well as in hippocampus-lesioned patients to examine the extent to which such functions are subserved by structures within the medial temporal lobes. Below are some sample references that complement the current behavioral findings and could lay foundation for how the current findings could relate to the underlying neural dynamics.
  • Emily Ruzich, Crespo-Garcia, Sarang S Dalal, Justin F Schneiderman (2019). Human Brain Mapping. Characterizing hippocampal dynamics with MEG: A systematic review and evidence‐based guidelines
  • Nuttida Rungratsameetaweemana & Larry Squire (2018). Learning & Memory. Preserved capacity for scene construction and shifts in perspective after hippocampal lesions.
  • Tom Hartley, Colin Lever, Neil Burgess, John O’Keefe (2014). Philos Trans R Soc Lond B Biol Sci. Space in the brain: how the hippocampal formation supports spatial cognition.

Reply: We have added a more thorough and in-depth discussion of the potential underlying neural mechanisms of the deficits and group differences we found in the current study (p. 11, lines 361-364 and lines 377, 383; p. 12, lines 396-399; p. 13, lines 462-466). We have also added the lack of neuroimaging data as a study limitation (p. 14, lines 482-484). We thank the reviewer for the helpful suggested references and added these to the Discussion.

Round 2

Reviewer 1 Report

The manuscript has been markedly improved after the revision. I have one major and some minor suggestions:

Major comment:

According to Table 1, some SCD participants were very young (less than 40 years). This group should have been better matched with MCI and AD dementia groups. Including so young participants may affect the results although age was a covariate in the parametric analyses. In addition, the nonparametric analyses of the VT trials could not be corrected for differences in age, which may seriously affect the results. This should be stated as a strong limitation of the study and the results of the differences between the groups, especially in the VT test, should be discussed and interpreted very carefully.

Minor comments:

Introduction
1) At the end of the first paragraph, the references 6 and 7 are not accurate as the papers do not report spatial memory complaints and issues with driving. The first study found navigation deficits in MCI and AD (dementia) patients using virtual reality and the second one detected events of topographic disorientation (TD) and those who had events of TD were disoriented outside the familiar territory. I suggest to add more accurate references of rephrase the sentence. In the last sence of this paragraph a reference is missing.

Results
1) 3.1. Demographics, use "patients with mild AD dementia" instead of "patients with AD"
2) 3.3. VT task, there is a missing word in the first sentence "...between groups on the number of correct (?) in the scene recognition task"
3) 3.4. Correlational analyses, r values of 0.4 and 0.33 are not "mediate to large" but weak to moderate. In addition, the r values do not match those in the abstract.

Table 1, Figure 3, Table 2 and Appendix A
1) Replace "AD" with "(mild) AD dementia"

Table 2
1) In Distance comparison, AD dementia patients seem to differ at  p < 0.05 and it is not indicated from which group. In addition, in the results, there is a statement that differences between AD dementia and SCD patients in this task did not survive correction for multiple comparisons 
2) Superscript “b” indicating significant difference with MCI is not present in the table

Author Response

Major comment:

According to Table 1, some SCD participants were very young (less than 40 years). This group should have been better matched with MCI and AD dementia groups. Including so young participants may affect the results although age was a covariate in the parametric analyses. In addition, the nonparametric analyses of the VT trials could not be corrected for differences in age, which may seriously affect the results. This should be stated as a strong limitation of the study and the results of the differences between the groups, especially in the VT test, should be discussed and interpreted very carefully.

Reply: Ideally, we would have a included a better age-matched group of SCD participants, but unfortunately this is inherent to working with (pre)clinical populations from a memory clinic. We agree with the reviewer that this should be stated as a limitation in the manuscript and have therefore added this to the Discussion (p. 13, lines 489-495).

Minor comments:

Introduction
1) At the end of the first paragraph, the references 6 and 7 are not accurate as the papers do not report spatial memory complaints and issues with driving. The first study found navigation deficits in MCI and AD (dementia) patients using virtual reality and the second one detected events of topographic disorientation (TD) and those who had events of TD were disoriented outside the familiar territory. I suggest to add more accurate references of rephrase the sentence. In the last sence of this paragraph a reference is missing.

Reply: we have rephrased and added the correct references.

Results
1) 3.1. Demographics, use "patients with mild AD dementia" instead of "patients with AD"
2) 3.3. VT task, there is a missing word in the first sentence "...between groups on the number of correct (?) in the scene recognition task"
3) 3.4. Correlational analyses, r values of 0.4 and 0.33 are not "mediate to large" but weak to moderate. In addition, the r values do not match those in the abstract.

Reply: we adjusted accordingly.

Table 1, Figure 3, Table 2 and Appendix A
1) Replace "AD" with "(mild) AD dementia"

Reply: we adjusted accordingly.

Table 2
1) In Distance comparison, AD dementia patients seem to differ at  p < 0.05 and it is not indicated from which group. In addition, in the results, there is a statement that differences between AD dementia and SCD patients in this task did not survive correction for multiple comparisons 

Reply: the reviewer is correct. We did not remove the * in the revision by mistake. We have adjusted accordingly.

2) Superscript “b” indicating significant difference with MCI is not present in the table

Reply: We have removed the superscript “b” from the legend as there were no differences between patients with MCI and AD dementia.

Reviewer 2 Report

I appreciate how carefully the authors have addressed all of my comments. Great work!

Author Response

We thank the reviewer for the thorough review and believe that addressing the comments raised by the reviewer significantly improved the manuscript